# Transformation of Waste Stabilization Ponds: Reengineering of an Obsolete Sewage Treatment System

Silvânia Lucas dos Santos [1] 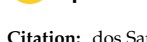 and Adrianus van Haandel [2],*

1 Department of Civil Engineering, Federal University of Rio Grande do Norte, 59.078-970 Natal, Brazil; silvania@ct.ufrn.br
2 Department of Civil Engineering, Federal University of Campina Grande, 59.429-350 Campina Grande, Brazil
* Correspondence: adrianusvh@gmail.com; Tel.: +55-83-99133-0196

**Abstract:** Waste Stabilization Ponds (WSPs) are commonly used for sewage treatment. These systems are composed of a series of ponds: (1) anaerobic ponds, (2) facultative ponds, and (3) maturation ponds. WSPs generally produce good-quality effluent in terms of organic matter and pathogen removal, but their application has disadvantages. The most serious disadvantages are a long retention time, the release of biogas, and the impossibility of removing nutrients. A promising alternative to the use of WSPs is replacing the anaerobic pond and facultative pond with an upflow anaerobic sludge blanket (UASB) reactor, with the advantages of greatly reducing the retention time and the biogas capture. The post-treatment ponds of the UASB reactor effluent involve oxygen production and the biological consumption of carbon dioxide, which raises the pH. An experimental investigation showed that it is possible to use polishing ponds in a sequential batch regime instead of continuous flow. This modification accelerates the decay of pathogens and accelerates the increase in pH, which, in turn, facilitates the removal of nitrogen and phosphorus. This produces a good-quality effluent with low concentrations of biodegradable organic material, nutrients, and pathogens. This good-quality effluent is obtained in a system without energy consumption or auxiliary materials and with a much smaller area than conventional stabilization ponds.

**Keywords:** waste stabilization ponds; UASB reactor; polishing ponds; sequential batch regime; investment costs



## 1. Introduction

At the beginning of the last century, when primary sewage treatment proved to be insufficient for the protection of water resources, secondary treatment options were developed to provide additional biological treatment for the removal of organic material. One of these methods was the waste stabilization pond (WSP) system, which was later improved to increase its treatment capacity. Parker et al. (1950) [1] consolidated the results of this experiment over the first half of the last century and developed the so-called Australian system, which showed that the ideal configuration was to subdivide the stabilization pond system into three sequential parts: (1) the anaerobic pond (AP), which receives the influent and is in an anaerobic condition; (2) the facultative pond (FP) that receives the effluent from the AP, which is at least partially aerobic because oxygen is generated due to algae-mediated photosynthesis; and (3) maturation ponds, which are predominantly aerobic, complement the removal of organic material, and improve the hygienic quality of the final effluent.

A second important development of the stabilization pond project was the recognition that the pond system also resulted in an improvement in the hygienic quality of the sewage. Marais and Shaw (1964) [2] showed that the decay of thermotolerant coliforms could be described as a first-order process. Later, Marais (1974) [3] developed a model to optimize stabilization ponds for the removal of thermotolerant coliforms. By joining the models

of Parker et al. (1950) and Marais (1974) [1,3], it is possible to design a stabilization pond system that efficiently removes both organic material and pathogenic organisms.

In the 1960s, it became evident that in addition to the removal of organic material and pathogens, the removal of the nutrients, nitrogen and phosphorus, is of vital importance to protect recipient bodies. However, the research aimed at removing nutrients by Pano and Middlebrooks (1982) [4], Bastos et al. (2018) [5], Zimmo et al. (2003) [6], and Camargo Valero and Mara (2010) [7] showed that nitrogen removal is, at best, partial, and the phosphorus removal is poor (see Gomez et al. (2000)) [8]. The reason for this failure is that the pH in a WSP remains close to the neutral point. In this article, it is shown that for efficient nutrient removal, it is necessary for the pH to increase significantly. However, in conventional WSPs, there is an equilibrium between oxygen and carbon dioxide production and consumption. As a result, the pH remains essentially constant, and nutrient removal is incomplete.

WSPs have some advantages, but they also have major drawbacks. The advantages of WSPs are associated with their operational simplicity and low cost of implementation (if there are favourable topographic conditions and the area is available at a low price). The disadvantages, however, are numerous and well-established, such as the following:

(1) Oxygen production is slow, so the WSP's area must be large (a required area of $3 \ m^2$/IE and total retention time of approximately 1 month, Mara (1976) [9]), to maintain at least a partially aerobic environment in the facultative pond and a predominantly aerobic environment in maturation ponds, which is essential for the efficient removal of organic material.

(2) The consequence of the large areas of WSPs is a considerable loss of water via evaporation, resulting in an increase in salinity of the effluent, thus impairing its applicability as water for irrigation purposes (Mara and Pearson (1998)) [10].

(3) WSPs are not suitable for the removal of nutrients (nitrogen and phosphorus), so effluent discharge can result in eutrophication of the receiving surface water body. Moreover, the WSPs' effluents with nutrients have little use as industrial water reuse.

(4) The application of AP results in the biogas production that emanates from the liquid phase and generates bad odours around the WSP system due to the presence of hydrogen sulphide gas, which greatly contributes to the unpopularity of WSPs with the surrounding population.

(5) Biogas from the AP represents an ecological problem: Methane in the biogas contributes significantly to the release of greenhouse gases and is 21 times more harmful than $CO_2$. As a result, WSPs have the largest GHG footprint of all treatment systems (Van Haandel and Van der Lubbe (2019)) [11].

(6) There is a significant accumulation of non-biodegradable solids in the AP as a result of settling, which leads to the need to remove solids (every 3–5 years)—a complicated operation (Cavalcanti (2003)) [12].

(7) The need to remove the pond and its odours from the urban region entails an extra cost for the collection network with a long outfall, representing a major investment factor.

(8) WSPs are almost invariably built as single treatment systems for cities, so segmentation of the sewage collection network is not possible, again leading to large investment costs for the sewerage network.

The treatment of WSPs is basically reduced to the removal of organic material and pathogens; for that reason, the effluent does not comply with legal standards in most cases. Specifically, nutrient removal is insufficient.

In view of these serious drawbacks of WSPs, a novel treatment system is proposed. This system comprises a combination of efficient anaerobic pre-treatment and post-treatment ponds. The most widely applied anaerobic sewage treatment unit is the upflow anaerobic sludge blanket (UASB) reactor, which on its own offers greater efficient BOD (Biological Oxygen Demand) removal than the AP and the FP combined [11] and produces a clear effluent.

## 2. Polishing Ponds as a Post-Treatment Alternative for Digested Sewage

As a result, in a subsequent post-treatment pond, photosynthesis is stimulated (with more transparency), and oxygen consumption is reduced (leading to less BOD). Therefore, photosynthesis tends to dominate over the oxidation of organic materials, leading to net oxygen production. This opens up the possibility of operating the post-treatment pond as a sequential batch unit, leading to a shorter retention time and decreased pond volume than treatment in flow-through units, thus reducing the pond area. Oxygen production is accompanied by carbon dioxide consumption, increasing the pH, which can lead to the removal of the nutrients, N and P.

Thus, the substitution of WSPs with a combination of a UASB reactor and post-treatment sequential batch ponds potentially offers several important advantages: (1) efficient BOD removal in the UASB reactor reduces the retention time in the subsequent pond; (2) capturing the biogas production avoids methane emissions and odour generation; (3) produced solids can be discharged from the UASB reactor; (4) the pH increase in post-treatment ponds allows the removal of nutrients via ammonia desorption and phosphate precipitation; and (5) since there is no odour problem, the novel system can be built near or even within urban areas, thus reducing construction costs. This paper's aim is to quantify these advantages.

Post-treatment after efficient anaerobic pre-treatment may be carried out in polishing ponds (PPs), which differ from the conventional stabilization ponds (WSPs) used for the treatment of raw sewage. Since UASB effluents are of better quality than raw sewage, the configuration and operation, as well as the objectives, of a PP will be quite different from those of a conventional WSP system for treating raw sewage.

Table 1 shows the differences between conventional stabilization ponds (WSPs) treating raw sewage and polishing ponds (PPs) treating effluents from the UASB reactor. The advantages of PPs are not limited to aspects of the treatment system itself (smaller area, less evaporation, lack of odour, use of sludge as fertilizer, possibility of nutrient removal, etc.) but also in terms of the sewage collection system (shorter outfall, the possibility of segmentation of the network in large cities, etc.). The reduced cost due to the shorter outfall and segmentation of the network is great, and often investment in the collection and treatment of sewage with a system composed of a network + UASB + PP may be smaller than the investment in the network alone for a conventional WSPs system due to a reduction in sewage network costs.

**Table 1.** Differences between conventional waste stabilization ponds (WSPs) and polishing ponds.

| Parameter | Waste Stabilization Pond | UASB + Polishing Pond |
|---|---|---|
| Influent | Raw sewage | Digested sewage |
| Main objective(s) | BOD and Pathogen removal | Pathogen and nutrient removal |
| Configuration | Series of ponds (AP-FP-MP) | Series of sequential batch polishing ponds (SBPPs), operated in parallel |
| Bottom sludge | Rapid accumulation in AP ($250\ mg \cdot L^{-1}$) | Slow accumulation, mostly algae ($70\ mg \cdot L^{-1}$) |
| Desired flow regime | Continuous flow with completely mixed ponds | Ponds with a sequential batch regime (SBPPs) |
| Retention time (in hot climate) | On the order of 1 month: organic material removal is limiting | Less than 1 week: thermotolerant coliform removal is limiting |
| Gas emissions | Biogas released to the atmosphere causing bad odours | Biogas is generated in the UASB reactor and can be flared off |
| Methane generation | Methane released contributes to GHG emission | GHG emissions are at the lowest possible level for sewage treatment |

**Table 1.** *Cont.*

| Parameter | Waste Stabilization Pond | UASB + Polishing Pond |
|---|---|---|
| Areas of application | Far from urban regions, requiring long outfalls | Proximity of the population is not a problem |
| Nutrient removal | Little (<20%) | Virtually complete N and P removal is feasible |
| Sewage treatment system | Centralized | "Segmented" network with separate treatment systems is feasible |
| Reuse | Limited to agricultural reuse | Reuse in agriculture and industry |
| Protection-receiving water body | No protection: Nutrients can cause extensive eutrophication | Almost complete nutrient removal is possible, avoiding eutrophication. |

Legend: AP = anaerobic pond; FP = facultative pond; maturation pond; UASB = upflow anaerobic sludge blanket; SBPP = sequencing batch polishing pond; BOD = biological oxygen demand; GHG = greenhouse gas.

### 2.1. Process Development in Polishing Ponds

In treatment ponds, several biochemical, physical, and chemical processes develop, some of which result in a reduction in the sewage's undesirable constituents. Figure 1 schematically shows the three biochemical processes that can develop and their products.

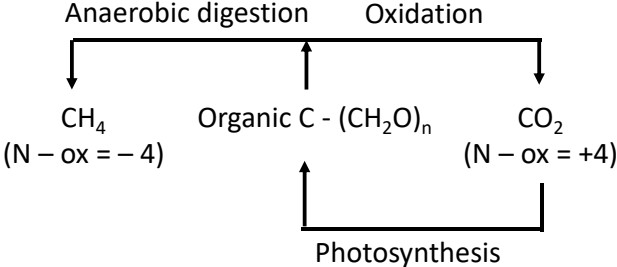

**Figure 1.** Schematic representation of biological processes in sewage treatment ponds.

(1) In the anaerobic environment (the AP and lower part of the FP), anaerobic digestion develops and results in biogas production from the digestion of organic material (OM):

$$OM \rightarrow CH_4 + CO_2 \qquad (1)$$

(2) In the aerobic environment, oxidation of the organic material occurs, which results in oxygen consumption and the generation of carbon dioxide:

$$OM + O_2 \rightarrow CO_2 + H_2O \qquad (2)$$

(3) At the same time, photosynthesis occurs, which is essentially the inverse process of oxidation involving the production of organic material and oxygen from carbon dioxide and water:

$$CO_2 + H_2O \rightarrow OM + O_2 \qquad (3)$$

There are also physical and chemical processes that develop in the pond parallel with the biochemical processes, as indicated in Figure 2. Physical processes are the processes of desorption or absorption of volatile compounds that are present in the UASB effluent or are generated in the ponds. These compounds are ammonia, carbon dioxide, dissolved oxygen (DO), methane, and hydrogen sulphide. In the case of $H_2S$, $NH_3$, and $CH_4$, desorption occurs because these gases have no appreciable concentration in the air. In the case of $CO_2$ and $O_2$, desorption or absorption may occur depending on whether the concentration of the compounds in the liquid phase is greater or smaller than the saturation concentration.

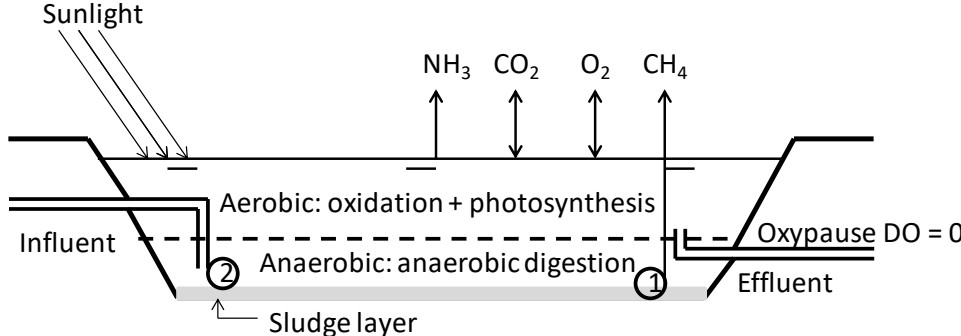

**Figure 2.** Schematic representation of physical processes that develop in sewage treatment ponds.

Absorption and desorption processes can be described by Fick's Equation (Equation (4)), which states that the desorption rate of a volatile compound in a liquid is proportional to the degree of supersaturation that exists between the current concentration of the compound and the concentration saturation:

$$r_d = k_d \, (C_l - C_s) \tag{4}$$

where $r_d$ is the desorption of the volatile compound; $k_d$ is the desorption constant; $C_s$ is the saturation concentration of the volatile compound; $C_l$ is the concentration of the volatile compound in the liquid phase.

Thus, the rates of transfer can be expressed as

$$r_{dN} = k_{dN} \, [NH_3] \tag{5}$$

$$r_{dO} = k_{dO} \, ([DO] - [DO]_s), \tag{6}$$

$$r_{dC} = k_{dC} \, ([CO_2] - [CO_2]_s) \tag{7}$$

The desorption of DO and $CH_4$ does not affect alkalinity or acidity; therefore, it also does not affect pH. $H_2S$ desorption, although noticeable, is very small (<0.5 meq/L) and also does not affect pH. Desorption of the acid $CO_2$ decreases acidity; therefore, it increases the pH. The desorption of ammonia increases the acidity and decreases the alkalinity; therefore, it tends to decrease the pH.

Other physical processes that develop in the ponds include the sedimentation of solids that can enter with the influent or form in the PP due to the flocculation of the algae growth. These solids are partially transformed into biogas (methane) via anaerobic digestion, with the unbiodegradable part accumulating at the bottom of the pond as a sludge layer. An important chemical process is the precipitation of phosphate salts. These salts only develop when there is a substantial pH increase in the pond, which can happen only in SBPP-type ponds. Under suitable conditions (pH ≈ 10), calcium carbonate can also precipitate.

### 2.2. Treatment Objectives of Polishing Ponds

The objectives of the PPs in treating effluent from a UASB reactor or other efficient anaerobic sewage digestion systems depend on the destination that will be given to the final effluent. For agricultural reuse, a reduction in the suspended solids and residual BOD and the efficient removal of helminth eggs and thermotolerant coliforms are important. In case of reuse of the final effluent in industry or its discharge into surface waters, additional nutrient removal is of fundamental importance.

The removal of thermotolerant coliforms (TTC) was described as a first-order process by Marais (1974) and Van Haandel and Van der Lubber (2019) [3,11]. Levenspiel (2003) [13] showed that, in this case, it is very advantageous to use reactors operating in a sequential batch regime. Depending on the depth, a value in the range of 3 to 7 days can be expected

for TCC removal, which entails a large decrease compared to the retention time of WSPs, which is approximately 30 d.

If the objectives of the PP include nutrient removal, the retention time in the PP will be longer because it will be necessary to increase the pH before ammonia desorption and subsequent phosphate precipitation.

## 3. Materials and Methods

The experimental systems were installed and monitored in the City of Campina Grande, Brazil. The utilized wastewater was raw municipal sewage from the city.

Polishing ponds:

The wastewater was treated anaerobically in a UASB reactor installed at the experimental site. The reactor has a volume of 2.5 m³ and a height of 1.7 m with a treatment capacity of 10 m³/d (Santos et al., 2016). After digestion, part of the UASB effluent was used to feed the polishing ponds.

The polishing ponds used in the experimental investigation were made of fibreglass, with a diameter of 0.5 m and depths of 0.2, 0.4, 0.6, and 1.0 m. Figure 3 shows a schematic representation and photo of the polishing ponds used in the investigation. Shallow ponds were used, as Cavalcanti (2003) showed that shallow ponds operate with high efficiency [12]. These pond models were operated with very gentle superficial stirring using a shallow superficial metal bar (1 cm width) attached to a small motor (6 rpm) to resuspend any algae floating within bubbles of dissolved oxygen that emerged from the ponds when they were supersaturated with DO. At the same time, the agitation served to even out stratification in the liquid phase. In practice, this gentle agitation might not be necessary for full-scale ponds because factors such as wind and sun-based thermal mixing could introduce enough mixing for a uniform liquid phase.

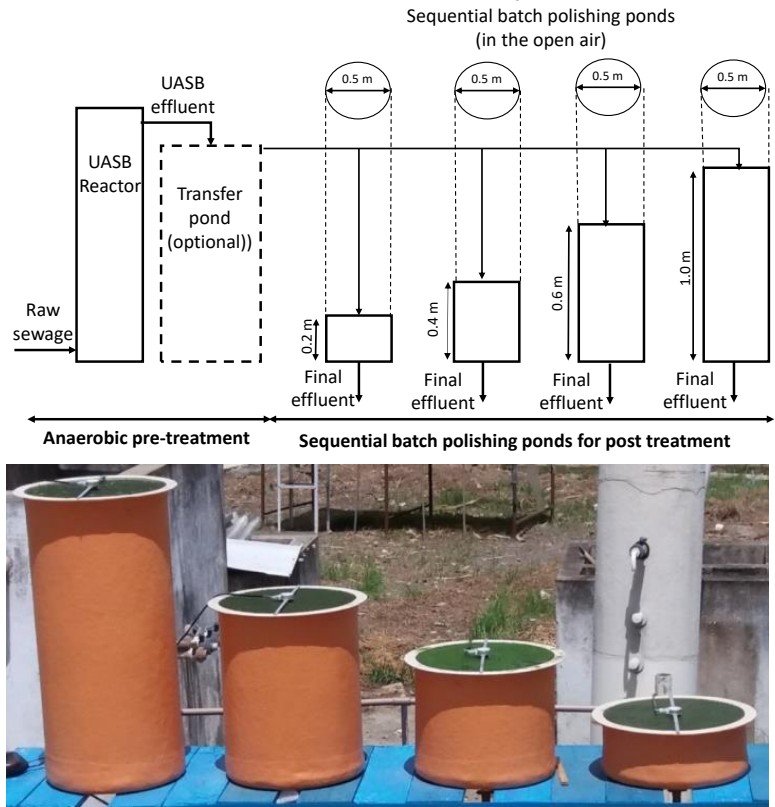

**Figure 3.** Flow sheet and photograph of an SBPP system with depths of 0.2, 0.4, 0.6, and 1.0 m with mild stirring of the pond contents.

The ponds were operated under a sequential batch regime since experiments developed by Albuquerque et al. (2020) showed that SBPP lagoons were more advantageous than CFPPs for all treatment objectives.

The experiments were conducted outdoors for a period of nine months. In Campina Grande, sunlight is abundant, and the sewage temperature remains at 25 °C, all year long. However, during winter, there is a greater incidence of rain.

Monitoring and evaluation of the system performance:

The SBPPs were operated without inocula. They were filled with effluent from the UASB reactor, and the operational variables of temperature, pH, dissolved oxygen (DO), biochemical oxygen demand (BOD), ammonia, and phosphate were assessed daily. Test completion was determined by two criteria: (i) when a phosphate concentration lower than 1 mg/L was reached or (ii) when an operation time of 30 d was reached.

Laboratory analyses were carried out to characterize the effluent of the UASB reactor and that of each of the lagoons. SBPPs were monitored daily since the operational regime caused rapid changes in the variables. Samples were always collected in the morning. Sampling was carried out by taking grab samples from the liquid phase. A total of 93 batches were carried out and distributed as follows:

L1 (0.2 m): 45 batches in summer and 11 in winter;

L2 (0.4 m): 30 batches in summer and 05 in winter;

L3 (0.6 m): 10 batches in summer and 03 in winter;

L4 (1.0 m): 08 batches in summer and 03 in winter.

The following variables were evaluated: DO, pH, temperature, BOD, COD (chemical oxygen demand), ammoniacal nitrogen, and phosphate. For the measurement of DO, temperature, and pH, a multiparametric probe (Hanna, model HI 98196) was used for online measurements, while for the other parameters, daily samples were analysed, following the procedures of the Standard Methods for the Examination of Water and Wastewater (APHA; AWWA; WEF, 2017).

## 4. Results and Discussion

Figure 4 shows the evolution as a function of time for the important SBPP variables: DO concentration, pH, COD, and ammonium and phosphate concentrations at different pond depths and under summer (left) and winter conditions (right) at Campina Grande. The figures show the trends of the variables and demonstrate the feasibility of applying the SBPP, at least under the climatic conditions of Campina Grande.

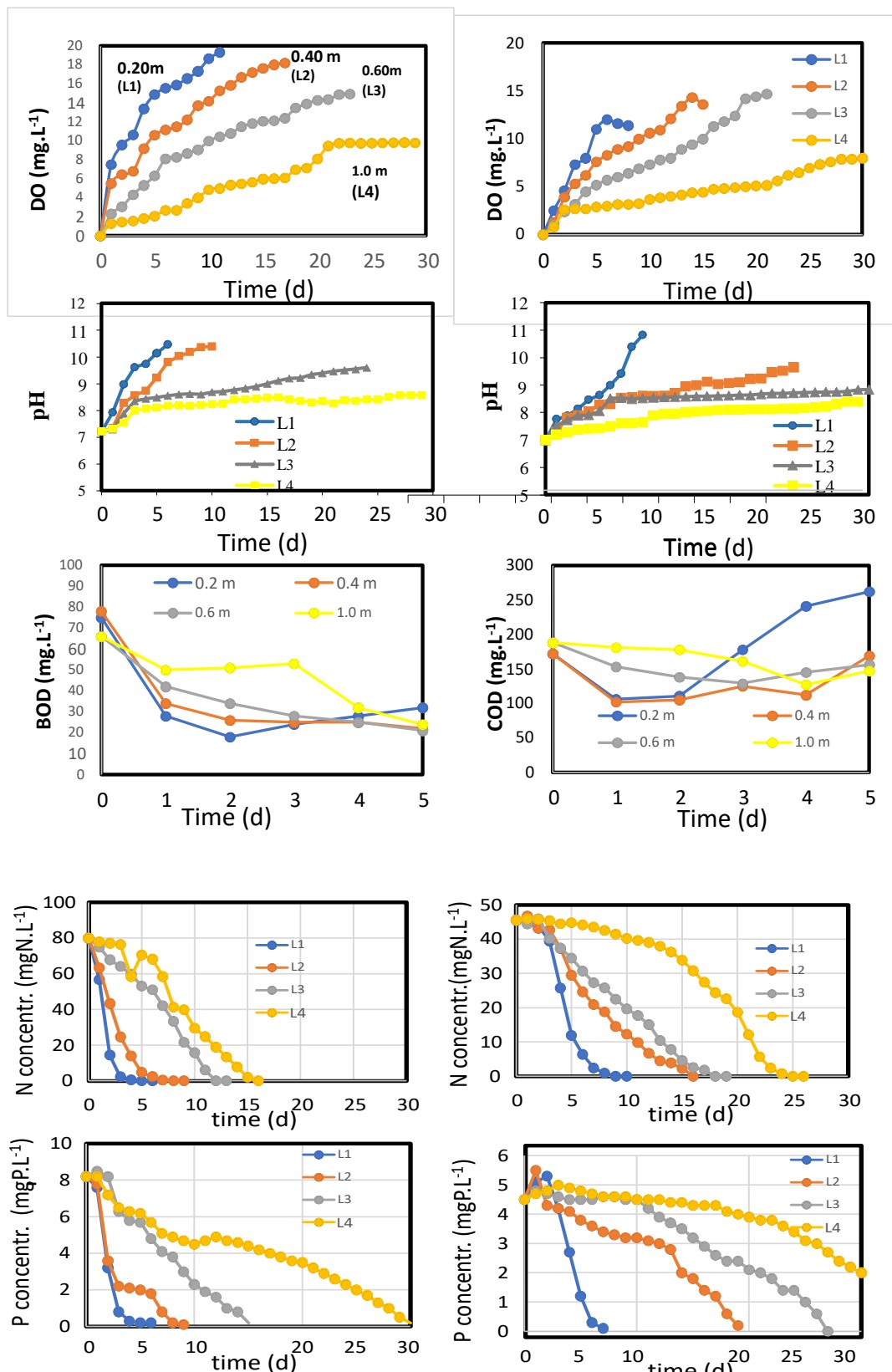

**Figure 4.** Experimental results of DO, pH, organic material, ammonium, and phosphate concentrations (L1 = 0.2 m; L2 = 0.4 m; L3 = 0.6 m; L4 = 1.0 m).

*4.1. DO Profiles*

Figure 4 shows typical profiles of the DO concentration as a function of batch time for different depths in the SBPP. Figure 4 illustrates the following:

(1) In all SBPPs, the DO concentration started at DO ≈ 0 and increased with time.

(2) For any retention time, the DO increased faster as the pond became shallower, which is expected given that photosynthetic production and oxygen transfer are faster in shallow ponds with relatively large areas.

(3) All profiles showed large daytime oscillations, which are attributable to the absence of photosynthetic DO production at night.

(4) The DO concentration trended towards its maximum and then decreased over time despite algae generation in the ponds, which may possibly be attributable to a decrease in activity due to a pH increase.

*4.2. pH Variation*

The processes that affect the pH variation in PP are photosynthesis and ammonia desorption [14]. Photosynthesis decreases acidity but does not affect alkalinity, so pH tends to increase. Ammonia desorption increases acidity and decreases alkalinity, resulting in an increase in pH.

As shown in Figure 4, the pH increased in all ponds over time. However, the rate of change depended on the depth: While in the SBPP with a 0.2 m depth, a pH above 9.5 was obtained in a few days, in the deeper ponds, this high pH was not reached after 30 d.

Due to the higher rate of increase in the pH in the shallow pond, it can be concluded that photosynthesis is faster, which is corroborated by the rapid increase in DO concentration. The reason for this difference is that in the shallow ponds, the area was relatively large, so more sunlight entered the liquid phase.

*4.3. Organic Material*

Figure 4 shows that the concentration of organic material can reach very low levels of BOD ≈ 20 mg/L and COD ≈ 120 mg/L. However, both BOD and COD increased with time, which can be attributed to the algae growth in the ponds, stimulated by the high transparency of the UASB effluent. In all cases, the increase in COD was much smaller than expected based on stoichiometry for the increase of oxygen in the pond (Equation (3)). This can be attributed to the algae flocculation and sedimentation in the pond. It can also be observed that the COD in shallow ponds (0.2 m) tended to be higher than the COD in deeper ponds.

*4.4. Ammonium Removal Efficiency*

According to Equation (5), the rate of ammonia desorption is proportional to the unionized ammonia concentration, which in turn depends on the total ammonia concentration and the pH:

$$NH_4^+ \leftrightarrow NH_3 + H^+$$
$$\text{or}$$
$$K_a = [NH_3]\,[H^+]/[NH_4^+] \tag{8}$$
$$\text{thus,}$$
$$[NH_3]/N_t = 1/(1 + 10^{(pH\text{-}pKa)})$$

Emerson (1975) [15] determined the dissociation constant $pK_a$ = 9.1 at 25 °C, which means that at pH = 9.1, the fraction of unionized ammonia is 50%, but at a neutral pH (pH = 7.1), the fraction is low (1%). Therefore, at a neutral pH, the rate of ammonia desorption is very low. As the pH increases, unionized ammonia and the desorption rate increase. The rate will decrease again when the total ammonia concentration becomes small.

*4.5. Phosphate Removal Efficiency*

The data in Figure 6 show that the SBPPs offered a high efficiency of P removal even under short retention times—5 and 10 days for ponds 0.20 and 0.40 m deep, respectively.

The phosphate removal efficiency is closely related to the pH value in the pond. In a subsequent paper, it will be shown that P removal was, in fact, due to hydroxyapatite precipitation and that for efficient P removal, a pH in the range of 9.5 to 9.7 is required. This high pH could not be reached in the FTPP (flow-through polishing pond), and for that reason, the phosphorus removal was poor (Cavalcanti (2003) [12] and Albuquerque et al. (2021)) [14].

Partial phosphate removal is not necessarily a problem for water reuse in industry. In this case, clarification or flotation is necessary to remove algae and other suspended solids. As ferric or aluminium salts are normally used, phosphate precipitation will occur in parallel and will leave a very low residual concentration of less than 0.1 mgP.L$^{-1}$. Naturally, the removal of suspended solids to produce a high-quality water industry generates the problem of determining what to do with the suspension that results from clarification. One solution is anaerobic digestion, where the digester could be the UASB reactor or a dedicated sludge digester.

*4.6. Thermotolerant Coliform Removal*

Several researchers have shown that the decay of thermotolerant coliforms in PPs is a first order process (Marais (1974) [3] and Van Haandel and Van der Lubbe [11]):

$$dN/dt = k_b N \tag{9}$$

where $N$ is the number of thermotolerant coliforms present at time $t$; $k_b$ is the decay constant.

Marais (1974) showed that the basic differential equation can be solved for different hydrodynamic reactors and proposed the following solutions for the removal efficiency of $N_e/N_i$ [3]:

(1)  Sequential batch reactor: An exponential relationship between the removal efficiency and the retention time:

$$N_e/N_i = \exp(-k_b HRT) \tag{10}$$

(2)  Complete mix continuous flow reactor: A hyperbolic relationship between the removal efficiency and the retention time:

$$N_e/N_i = 1/(1 + k_b HRT) \tag{11}$$

(3)  Complete mix series reactor systems: For a series N of equal reactors, one has:

$$N_e/N_i = 1/(1 + k_b HRT/N)^N \tag{12}$$

These solutions indicate the following:

(1)  For any retention time, the most efficient solution for bacterial removal is to operate a sequential batch reactor;
(2)  A series of complete mix flow through the reactors is more efficient than a single reactor with an equal volume, and this difference increases with the number of reactors in the series.

In Figure 5, the ratio of the retention times in a completely mixed flow series through ponds and a sequential batch polishing pond is plotted as a function of the number of ponds in the series. The difference in retention time (and, hence, in the pond area) is very large, especially if the number of ponds is not great. Thus, for a removal efficiency of 99.99% of thermotolerant coliforms, the retention time in a series of four FTPP is 4 times longer than that in an SBPP, which means that the size of each of the four flows through the ponds is equal to that of the single sequential batch pond.

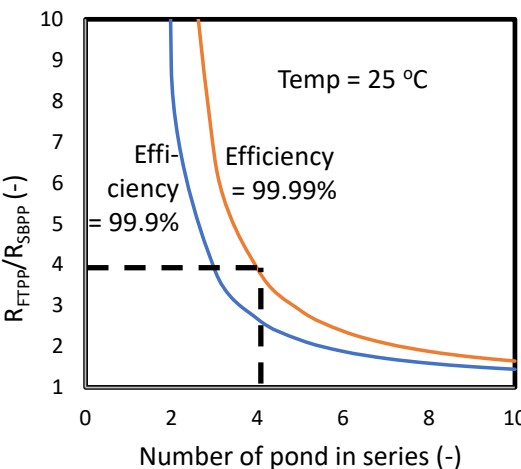

**Figure 5.** Ratio of retention times in a series of FTPPs (complete mix) and in an SBPP for removal efficiencies of 99.9% (blue curve) and 99.99% of the influent thermotolerant coliforms (red curve) as a function of the number of FTPPs.

It is concluded that the operation of the SBPP is advantageous and reduces the size of the post-treatment unit significantly.

In practice, the most important variable for pond design is the required area for treatment. This area is related to the retention time, as shown by Equation (13):

$$A_{IE} = V_{IE}/H = q_{IE}R_h/H \qquad (13)$$

where $A_{IE}$ and $V_{IE}$ are the area and volume of the polishing pond per inhabitant equivalent (IE), respectively; $q_{IE}$ is the contribution per inhabitant equivalent; $H$ is the pond depth; $R_h$ is the retention time in the pond.

The data in Figure 3 and Equation (13) were used to construct Figure 6, in which the required area per inhabitant equivalent for the different treatments (removal of COD, nitrogen, and phosphorus) is plotted as a function of the pond depth for summer and winter conditions at Campina Grande (7° South). Additionally, the per capita area for TTC removal is plotted in Figure 6 (Equation (10)). A decay constant of $k_b = 1.6/H*1.07^{(t-20)}$ was determined by Medeiros et al. (2021) [16] for the conditions in Campina Grande. The Figure assumes a per capita contribution of 100 L/d but can be adapted for any other value since the area is proportional to the contribution.

The results plotted in Figure 6 indicate the following:

(1)  The area required for SBPP to remove BOD and coliforms is much smaller than the value required for conventional stabilization ponds, which is about 3 m$^2$ per inhabitant. The data indicate that with SBPPs, for the BOD and thermotolerant coliforms (TTCs), the area is reduced by a factor of 4 to 5 compared to that of conventional WSPs.

(2)  If nitrogen removal is required, the SBPP area is reduced by a factor of 2, approximately.

(3)  If phosphorus removal is desired, the SBPP area is more or less equivalent to that of a WSP;

(4)  The SBPP area for organic material and CTT removal is only marginally affected by climate and depth. In contrast, the area for removing nutrients depends on these factors;

(5)  The depth of SBPP has an optimal value of about 0.5 m for nutrient removal, which is a value much smaller than that normally applied to WSPs;

(6)  The removal of phosphorus requires about twice the area required for the removal of ammonia.

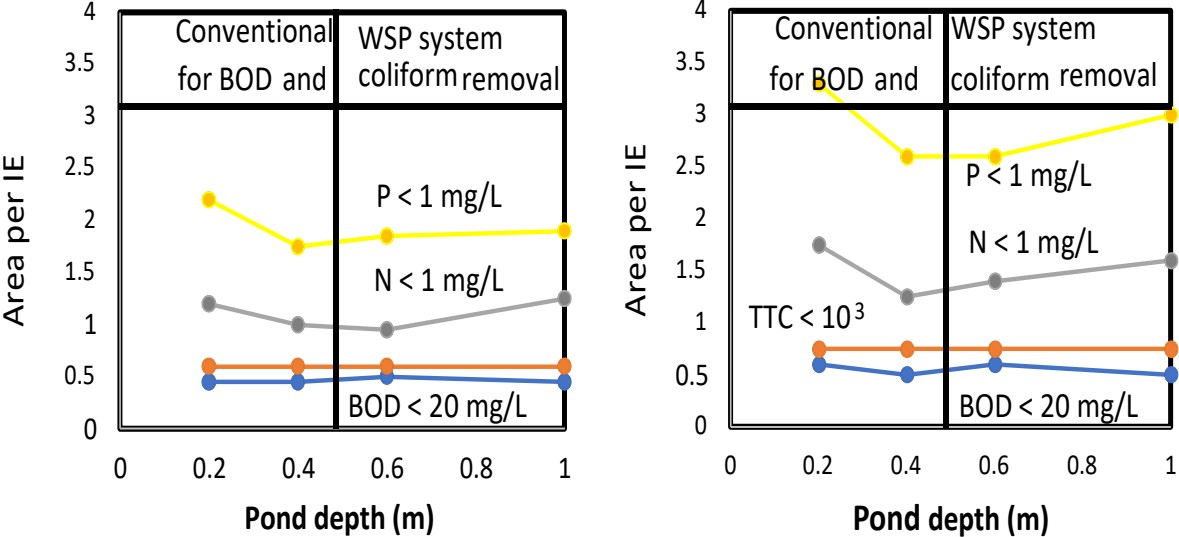

**Figure 6.** Area per inhabitant equivalent SBPP for the removal of BOD, thermotolerant coliforms, nitrogen, and phosphorus as functions of the pond depth under summer (left) and winter conditions (right) in the tropics (per capita contribution of 100 L·d$^{-1}$).

### 4.7. Results of the Experimental Investigation

The most important results of the experimental investigation can be resumed as follows:

(1) SBPP can be operated with a much shallower depth than conventional maturation ponds (MPs) (Figure 4).
(2) The area required for SBPP is much smaller than conventional MPs (Figure 6).
(3) There is a considerable advantage in using SBPP rather than FTPP.
(4) Foul odours are avoided in SBPP; thus, they can be applied near urban regions.

### 4.8. Reengineering of Waste Stabilization Pond Systems

The presented data show that the inclusion of a UASB reactor as a pre-treatment unit enables a radical modification in the size, operation, and configuration of the ponds as post-treatment units, taking into account the gaps still present in pond research, especially with regard to biochemical aspects (Espinoza et al., 2020 [17]; Verbijla et al., 2015 [18]; Ho et al., 2019 [19]). The traditional flow-through ponds can be substituted with great advantage via sequential batch ponds, which have a much smaller area and allow significant improvements in final effluent quality, with the possibility of efficient nutrient removal in the ponds. The area required for the SBPP depends on the SBPP's function—the removal of residual organic material, pathogens, or nutrients. There is an optimal depth in the range of 0.4 to 0.6 m for the SBPP that would allow operation of the smallest per-capita area for SBPP. This value is much smaller than the depths used in WSP maturation ponds, which are usually 1–1.5 m.

The SBPP produces final effluent quality equal to the WSPs in terms of the BOD and TTC removal efficiency but in a much smaller area. The SBPP has the great advantage of offering the possibility of removing nitrogen and phosphorus, which cannot be achieved in WSPs.

For the configuration, the conventional system of ponds in series and continuous flow (AP + FP + MP) is replaced by a series of PPs operating in parallel independent of each other under a sequential batch regime. This regime is possible thanks to the low concentration of biodegradable material in the UASB reactor effluent. It was shown that for all studied depths (range 0.2 to 1.0 m), the environment in the SBPP was always aerobic, and the DO concentration rapidly increased from a value of about 0 mg/L at the start of the batch to a value above the saturation concentration in the shallow ponds (0.2 to 0.4 m).

Having defined the area and depth of the UASB–SBPP system, it is necessary to determine the number of ponds to be constructed. There are essentially two possible configurations [11]:

(1)   A series of ponds receiving the UASB effluent sequentially in each pond (Figure 7a).

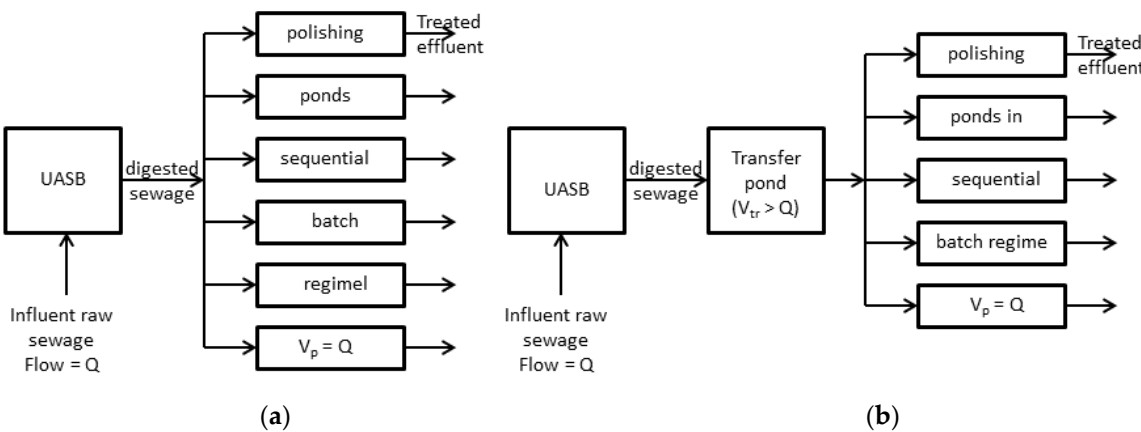

**Figure 7.** Schematic representation of the flow sheet and operation of polishing ponds fed with sequential batches without (left, (**a**)) and with (right, (**b**)) an intermediate transfer pond.

In this configuration, the retention time for the TTC decay is counted from the moment that the UASB effluent fills the pond and starts to feed the next one in the series. For operational convenience, one can choose the volume of the pond such that the time to fill the pond is one day. Thus, the total time spent in the pond will be the filling time (1 day) plus the retention time for the chosen objective of the SBPP. The number of ponds for the treatment is the retention time plus the time to fill the pond.

(2)   A series of ponds receiving the effluent from an intermediate transfer pond between the UASB and LPBS (Figure 7b).

In this configuration, the transfer pond has the function of an equalization tank, which receives a continuous flow of effluent from the UASB while the batches are discharged into the SBPPs, allowing for the sequential batch regime. The transfer pond is not only an equalization tank but also functions as a settler for settleable solids that will eventually be discharged from the UASB effluent, which may be a fraction of 30–50% of the produced solids [11]. If photosynthesis is sufficient in the transfer pond, sulphide removal may be possible if the oxygen production rate is compatible with the sulphide load on the transfer pond. In this way, it becomes possible to remove sulphate in industrial wastewater by reducing it to sulphide in the UASB reactor, followed by the oxidation of sulphide to sulphur in the transfer pond.

*4.9. Applications of the Proposed Sewage Treatment System*

The most obvious and immediate application of the proposed UASB–SBPP system is the transformation of the numerous conventional stabilization pond systems into systems with efficient pre-treatment (UASB reactors) followed by sequential batch-polishing ponds. Not only will the population surrounding the WSP systems benefit by removing uncomfortable odours, but there will also be significant environmental gains, such as burning biogas instead of releasing it into the atmosphere and the possibility of protecting the receiving bodies, thereby avoiding eutrophication. In regions with scarce water resources, one of the most important improvements yielded by the new treatment system is the possibility of generating a new source of water for use in industries, thus reducing the demand for public supply water and unlocking the chains of economic development caused by a lack of water. All benefits that can be materialized via the application of the novel system are in

line with the Sustainable Development Goals (SDGs), which will undoubtedly contribute significantly to a more sustainable environment, Ho et al. (2017) [20].

Systems composed of UASB–SBPP reactors can be much cheaper than conventional WSP systems for three reasons: (1) they can be built close to urban regions (or even within these regions), thus reducing the costs of the final outfall; (2) they can reduce costs due to a reduction in the area needed for implementing the system and in the height of the slopes of the ponds, which will be much smaller; (3) as the various SBPPs operate independently of each other, they can be built at different levels on terrain with rugged topography, thus reducing earthmoving costs.

The UASB–SBPP system is applicable at any scale, although it will generally not be used in large urban agglomerations because it requires a considerable area, despite being much smaller than that of a conventional WSP system. This system can also be applied to small flows. In rural areas, in the absence of a sewage collection network, the possibility exists to build single-family systems that reuse the effluent for food production on the properties.

### 5. Conclusions

(1)  A novel sewage treatment system composed of a UASB reactor and a series of sequential batch polishing ponds was proposed (UASB–SBPP) as a substitute for the conventional waste stabilization pond system (WSPs). The most important advantages of the new system are (1) the possibility to remove nitrogen and phosphorus, (2) a reduction of the required area, and (3) elimination of the vile odours emitted by collecting and burning the biogas.

(2)  Treatment in WSPs is limited to a reduction of organic materials and thermotolerant coliforms such that the effluent quality is not compatible with legal standards and, strictly speaking, cannot be discharged into surface waters. The UASB–SBPP system also removes organic material and thermotolerant coliforms but does so much more efficiently and is also able to remove the nutrients, nitrogen, and phosphorus.

(3)  The predominance of photosynthesis over the oxidation of organic material in polishing ponds leads to the consumption of carbon dioxide and increases pH. This high pH permits the removal of nitrogen due to the desorption of ammonia and phosphorus due to the precipitation of phosphate. In this way, a final effluent quality compatible with legal standards can be produced.

(4)  If the UASB–SBPP system is used only for organic material and coliform removal, the area is reduced by a factor of 4–5 compared to the WSP system. If nitrogen removal is also an objective, the area is reduced by a factor of 2 compared to the WSP system.

(5)  Costs of the UASB–SBPP are reduced because there is no need for a long outfall; further, the polishing ponds are much smaller and shallower and do not need to be levelled as they operate independently.

(6)  Due to the virtual absence of odour problems, the UASB–SBPP system can be constructed near or even within urban regions, avoiding high costs for the collection system. Moreover, the steep reduction of the PP area itself leads to an important cost reduction and can augment the system's applicability. Another factor related to cost reduction is that the pond depth of PP is much smaller than that of WSP, which reduces excavation costs.

(7)  Transfer ponds function not only as equalization tanks but also act as settlers, facilitate the retention of helminth eggs, and enable $CO_2$ desorption and sulphide's oxidation to sulphur.

**Author Contributions:** A.v.H.: Conceptualization, Methodology, Resources, Writing. S.L.d.S.: Validation, formal analysis, investigation, supervision. All authors have read and agreed to the published version of the manuscript.

**Funding:** The authors thank the support of the National Research Council (CNPq).

**Conflicts of Interest:** The authors declare no conflict of interest.

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
