# Peer review of "Transformation of Waste Stabilization Ponds: Reengineering of an Obsolete Sewage Treatment System"

_water, doi:10.3390/w13091193_

Round 1

Reviewer 1 Report

I enclose comments and suggestions for Authors

Author Response

A revised manuscript was produced taking into account the comments of the three reviewers and sent separately .

Reviewer 2 Report

Authors present a work on the use of polishing ponds for water treatment. Sadly, the manuscript is very dense, making difficult to read it. Furthermore, there are lots of spelling and grammar mistakes. The structure of the article is not adequate and it lacks of the rigor needed to be published in Waters. 

If you consider submission to another journal you should take into account the following issues:

  1. English should be revised throughout the text. 
  2. "(b) Facultative pond (FP) receiving the effluent from the LA" Please define all abbreviations prior to their use. In this case, what does LA stand for?
  3. When introducing the work of an specific group it is recommended to describe it this way: "Wang et al. studied the use of ponds [1]." Instead of "[1] studied the use of ponds". Please revise this format issue throughout the text. 
  4. Figure 6, one of the graphs has the axis title in Portuguese, please uniformize this.  Furthermore, the shade of yellow employed for BOD and pH is hard to see, please use the same tone as the one employed in the rest of graphs in that same figure. 
  5. Table 1, I would not say that effluents can be reused for public water supply as water reuse legislations in each country are very different. 
  6. Line 198, please eliminate the ! sign after eutrophication. 
  7. Line 199, please define what OD stands for. 
  8. You have an unusually large introduction section and a very short materials and methods section. Please introduce the analytical procedures employed to follow the BOD, N, P and pathogens. 
  9. Furthermore, it seems you have a larger introduction than even results. The article structure should be revised and adapted to that of a scientific article.
  10. DO or OD? You use both terms indisticntly without defining any of them. 
  11. Authors should improve the explanation of the reactors/ponds configuration. Does sequential (SBPP) mean that the UASB effluent goes through L1 to L2, L3 and L4? A diagram explaining the different configurations would be very useful.  
  12. What does LPFC and LPBS stand for? You should be more careful and revise all abbreviations. 
  13. Figure 6. It is season. Seazon does not exist
  14. Conclusions should be based on your results. You present a series of conclusions for WSP which are based on a bibliographic review and do not correspond to your work. 

Author Response

A revised manuscript was elaborated and sent separately 

Reviewer 3 Report

Efficient removal of organic matter, nutrients and pathogens is of paramount importance in the light of water bodies eutrophication and degradation. Densely populated urban areas are usually provided with sewage system and waste water treatment plants, while it is economically unsound on areas with dispersed small villages. Therefore, other systems, based on ponds is developed. There are multiple issues when coming to its exploitation, including odours, not enough efficient purification or high costs of land designed for ponds, as it was clearly explained by manuscript authors. Thus, any solutions dealing with those problems are expected.

The effectiveness of one of them was presented in the manuscript, however the numerous deficiencies make this proposal unclear and difficult to assess.

Firstly, the structure of the manuscript is disturbed. Very long introduction presents too many data, including tables and figures, usually absent in this chapter. Material and methods on the other side is definitely too scarce. Conclusions are made in ‘results and discussion’, while in the ‘Conlusions’ information already presented in ‘Introduction’ appears. The number of references is quite short. Severe revision is required in this matter.

Secondly, there is no clear study goal in the paper. It shall be explained in the end of Introduction. At this part, I am not sure what problems are addressed by Authors, as many advantages of studied purification system are indicated in Introduction.

As it was already mentioned, ‘Material and methods’ are absolutely scarce There is no chance of repetition of presented manuscript, what is required when it comes to this part of manuscript. No data on analyses made and methods used. How long the experiment last, how frequently were the samples collected? What were the dimensions of the experiment system (more data than depth only needed)? What was the difference between FTPP and SBPP in the experiment? Poor quality photo does not explain anything, a scheme would be much better. How many repetitions was used in each experiment? Was the experiment outside, if so, what about weather conditions? At this moment, the ‘Material and methods’ are not acceptable.

Finally, it seems to me that too many information from handbooks was included in the manuscript. Its amount is disproportionate to the results obtained by Authors.

When is comes to the technical aspect of the manuscript, in my opinion there shall be no column division to fit the figures. The manuscript requires thorough review when it comes to used acronyms. There is lot of them, not all are explained, lot of them are used in different forms (TCC, CTT, DO, OD etc.). Additionally, there is some incorrect numbering of figures in the text.

Other comments:

Line 6-9 – the affiliation shall not include the position of the authors, same name of institution is enough

Line 38 – ‘by Parker et al. [1], who developed’

Line 45 – what does the acronym LA stands for? Probably ‘anaerobic lagoon’, but this pond was named AP in previous paragraph

Figure 1 – no legend for acronyms (OD, inh, hab) – not all readers will recognize them as oxygen content (DO is usually used) or inhabitants

Line 67 – ‘Marais and Shaw [6] showed that…’, same for citations in lines 68 and 69

Line 127 – what does acronym UASB stands for?

Figure 2 – no legend explaining the acronyms used; what is the unit for HRT in UASB? Litres?

Table 1 – according to figure 1, UASB receives raw sewage, while PP gets digested sewage, so if UASB +PP is mentioned in the table, raw sewage are under treatment; unit is mg L-1, not mg.L-1;

Line 253 – CTT needs explanation

Line 282-283 – do not use such statements ‘Figure shows, figure permits’, describe the results

Line 307 - ‘both BOD and COD’

Figure 6 – what does L1, L2… stands for? Correct the legends – not the same on all figures, remove Spanish words; use the same font on all figures; why there is one figure for BOD and one for COD?

Line 319 – what does LPFC and LPBS stands for? I do not see the results for ammonia on figure 9.

Figure 7 – what is the point to place a graph coming from a handbook? ammonia transformations are well recognized, thus obtained results might be explained with 2-3 phrases, without equations and figures

Line 326 – no figure 11

Line 339 – 0.1 mgP L-1 is in fact quite high when it comes to lake ecosystems, stimulating the phytoplankton blooms

Figure 8 – name the blue and red lines on the graph or in figure caption

Line 378 – unify the acronyms TCC and CTT

Figure 9 – why there is information about conventional WSP system on the graph? Is this the comparison of SBPP efficiency in two seasons (as indicated in the text and figure caption) or comparison of SBPP and WSP?

Line 393 – 416 - seems more like conclusions of already presented results, suggest to remove this part

Lines 472- 492 – repetition from introduction, definitely remove, this is not the summary of your work but summary of literature

Lines 527-533 - remove

Author Response

A revised manuscipt was elaborated and sent separatly

Round 2

Reviewer 2 Report

Authors present a new manuscript with vague changes in relation to the first one. These modifications have not been marked in the manuscript, being difficult to adress the changes. Furthermore, the response to reviewer presented (at least my response) is a mix of portuguese and English in which they have not bothered to answer some of the claims and the majority of responses are "ok" or "done".

The structure of the article is still decompensated with a very large introduction and short materials and methods section, where no analytical methods are explained. 

Conclusions are too large, where most of them could be conclusions of a literature revision rather than a research paper. 

Once again, I sincerely believe that this manucript lacks the scientific rigor to be published in this journal and should be addressed elsewhere after a complete rewriting. 

Author Response

Answers to comments/suggestions by reviewer 2

(1) We have tried our utmost to introduce the answers to the comments and suggestions of the reviewers and this has led to considerable changes in the manuscript. 

We are unaware of any Portuguese text in the answers, but if there was, we apologize. 

The  "ok" and "done" answers are there only to indicate that all have been   corrected in the text

(2) We have reduced the introduction and extended the section of materials and methods. There is a section between "introduction" and "materials and methods" (Polishing ponds as a post treatment alternative for digested sewage), where we explain the how polishing ponds work, since that may not be known to many readers. 

The analytical methods used in the paper are all simple tests and one expects that these are well known to the reader and do not need more explanations.  

(3) The conclusions have been rewritten and have become much smaller. Only conclusions directly related to the presented research are presented. 

(4) The very fact that we are sending a new version of the paper indicates that we do not agree with this comment

Reviewer 3 Report

I find the manuscript improved due to implemented corrections. I especially appreciate the details in ‘Material and methods’, including the scheme of experimental design.

However, in my opinion the manuscript still requires some changes to be accepted for publication.

First of all, the ‘Introduction’ remains almost the same, with only minor changes in comparison to first draft. It is definitely too long and detailed. The aim of the study is still not clearly indicated in this chapter. In fact, it appears in the first line of ‘Material and methods’. The idea of ‘Introduction’ is to present what is already known in particluar matter and what are the blanks to be filled with study presented in following part of the manuscript. At this moment, the amount of information in ‘Introduction’ makes an impression that the proces of sewage purification is well recognized, and the blanks are not clearly explained.

Moreover, there is still some confusion when it comes to acronyms. Generally, when you use acronym for the first time in the text, you unravel it. It applies to all abbreaviations used in the text, tables, figures. Scientific manuscripts are read also by students, and we can not expect from them to know all acronyms. Therefore, in my opinion UASB shall be explained. Aditionally, the are still portugese acronyms (line 455) in the text, and part of acronyms are not explained (see Figure 2).

Figure 6 has been improved, however BOD and COD are not the same as others. Decide whether you want to describe the lines on graphs, or in figure caption – there is no need to use both solutions at the same time.

There is no figure 7.

BOD is not mentioned among studied parameters in ‘Material and methods’.

Author Response

(1) However, in my opinion the manuscript still requires some changes to be accepted for publication.

We have made extensive modifications in the section "Introduction" and "Materials and methods". "Introduction" has been reduced and Materia and methods has been increeased in accordance with you suggestions. Between these two section there is "Polishing ponds as a post treatment alternative for digested sewage" to explain how polishing ponds work, since this may not be clear to the reader. 

The objective of the paper is now at the end  of the introduction. 

(2)Moreover, there is still some confusion when it comes to acronyms. Generally, when you use acronym for the first time in the text, you unravel it. It applies to all abbreaviations used in the text, tables, figures. Scientific manuscripts are read also by students, and we can not expect from them to know all acronyms. Therefore, in my opinion UASB shall be explained. Aditionally, the are still portugese acronyms (line 455) in the text, and part of acronyms are not explained (see Figure 2). 

Hopefully all confusion of outlandish anacryms is now corrected and UASB has been duely explained

(3)Figure 6 has been improved, however BOD and COD are not the same as others. Decide whether you want to describe the lines on graphs, or in figure caption – there is no need to use both solutions at the same time. 

The Figures for BOD and COD are difeerent for two reasons: (1) the retention time of these is much shorter than the other ones and (2) after an initial decrease the BOD and COD values tend to increase, which is not the case with the others. 

(4) There is no figure 7

Indeed there is no figure  7

BOD is not mentioned among studied parameters in ‘Material and methods’.

BOD is mentioned on page 253

Round 3

Reviewer 2 Report

Authors present a third manuscript which still presents basic typing mistakes which should have already been polished. 

  1. Line 216: decimal separator should be a . instead of , (1.7 m instead of 1,7 m)
  2. Line 217: Reference should not be all in caps. 
  3. List of reagents and suppliers should be included in the materials and methods section. 
  4. Figure 4. Units in the Y axis of severals graphs should be revised and -1 put in subscript. Color of the axis numbers and letter sized uniformized (some are black, other gray)
  5. Line 320-322 revise again the use of comma for decimal numbers. Same in Line 327. Revise throughout the text, this error is spread throughout the whole manuscript.
  6. Conclusion number 6 is missing 

Author Response

Queries 1, 2, 4  and 5 have been changed conforming instructions.

Query 3 refer to elementary chemical tests that are described in Standard Methods (in the reference list). I have never seen a paper, where the details of these tests are mentioned

Query 6: conclusion 6 is there. 
